# Angiotensin II Receptor Blockers in the Management of Hypertension in Preventing Cognitive Impairment and Dementia—A Systematic Review

**DOI:** 10.3390/pharmaceutics14102123

**Published:** 2022-10-06

**Authors:** Elvira D’Silva, Nur Farah Meor Azlan, Jinwei Zhang

**Affiliations:** 1Hatherly Laboratories, Institute of Biomedical and Clinical Sciences, Medical School, Faculty of Health and Life Sciences, Streatham Campus, University of Exeter, Exeter EX4 4PS, UK; 2School of Medicine, Xiamen Cardiovascular Hospital of Xiamen University, Xiamen University, Xiang’an Nan Lu, Xiamen 361102, China; 3State Key Laboratory of Bioorganic & Natural Products Chemistry, Research Center of Chemical Kinomics, Shanghai Institute of Organic Chemistry, Chinese Academy of Sciences, # 345 Lingling Road, Shanghai 200032, China

**Keywords:** hypertension, dementia, systemic review, antihypertensive

## Abstract

Hypertension is a known risk factor for cognition-related pathologies including dementia. The National Institute of Health and Care Excellence (NICE) guidelines recommend angiotensin (Ang) II receptor blockers (ARBs) or angiotensin-converting enzyme inhibitors (ACEIs) as a first-line treatment for hypertension. Although both ARBs and ACEIs show neuroprotective effects, ACEIs show contradictory side effects; therefore, ARBs may be a more viable option. However, trials assessing the effects of ARBs on cognition are scarce and conflicting. Therefore, the aim of this review is to conduct a systematic review and synthesise data on the influence of ARBs on cognition and dementia prevention. Five databases were searched from 1992–2022 to produce 13 randomised controlled trials (RCTs) involving 26,907 patients that compared associations of ARBs against placebos or other antihypertensives on cognition or probable dementia with a minimum duration of 3 months. ARBs showed greater cognitive benefits when compared to hydrochlorothiazide (HCTZ), beta blockers (BB), and ACEIs. Our findings showed that although ARBs are superior to some antihypertensives such as ACEIs, thiazide and beta blockers, they made no difference in comparison to the placebo in all but one sample of patients. The positive effects on cognitive performances are equal to calcium channel blockers (CCBs) and lower than statin. The neuroprotective effects of ARBs are also more beneficial when ARBs are taken at the same time as a statin. Due to these inconsistencies, robust conclusions cannot be made. Future trials are warranted and, if successful, could have positive economic implications and consequently improve quality of life.

## 1. Introduction

Dementia is not a single condition but rather a heterogeneous disorder—a grouping of symptoms that occur together and form a recognised pattern (Table 1). Dementia typically affects an older population and is known to result in a lower quality of life due to its impact on memory, behaviour, thinking, and social abilities [1,2]. Currently, around 55 million individuals worldwide suffer from dementia, and this figure is predicted to increase significantly over the next decade [3]. Although the causes of dementia are unclear, the consequences are structural and chemical alterations in the brain. This can progress to neuronal death and brain volume reduction [2].

Although no curative treatments are available, treatment of several risk factors may delay the onset of dementia [4]. Among the cardiovascular risk factors, one of the key factors is high blood pressure or hypertension. Several studies have demonstrated that hypertension is associated with increased risk of developing dementia, particularly vascular dementia, the second most common type of dementia after Alzheimer’s disease [5]. Due to the high prevalence of hypertension and widely effective medical treatments, optimising antihypertensive treatment might potentially be a low-cost and scalable strategy to reduce dementia incidence.

Among the many antihypertensives available, the National Institute of Health and Care Excellence (NICE) recommends treating hypertension with an angiotensin II receptor blocker (ARB) or angiotensin-converting enzyme inhibitor (ACEI) as a first-line treatment for hypertension [6]. Both ARBs and ACEIs reduce blood pressure through inhibiting elements of the renin–angiotensin–aldosterone system (RAAS). Upon activation by factors such as a drop in blood pressure, renin is secreted into the bloodstream. Renin then acts on angiotensinogen to produce angiotensin I. Angiotensin-converting enzyme (ACE) then works on angiotensin I to form angiotensin II. Angiotensin II stimulates the secretion of aldosterone, which is involved in the retention of sodium. Retention of sodium leads to increased blood volume and therefore increased blood pressure. ACEIs work through inhibiting ACE from converting angiotensin I to angiotensin II. ARBs work though inhibiting the binding of angiotensin II to angiotensin receptors and have been shown to be neuroprotective at a central and peripheral level. The proposed molecular mechanism of ARB neuroprotection includes reducing angiotensin II toxicity, reduction of apoptosis and neurotoxic inflammation, and activation of PPARy. These mechanisms were found through studies using primary cultures of brain cells and animal models of diseases. One example of neuroprotective effects of ARBs is in the context of Parkinson ’s disease, where ARBs protect dopaminergic cells against injury [7]. An extensive discussion of the neuroprotective effects of ARBs and their possible molecular mechanisms can be found in the review by Villapol and Saavedra [8].

Although previous studies have shown that antihypertensive drugs may be associated with reducing cognitive impairment, current evidence for the superiority of ARBs is contradictory. Two observational studies indicated that ARB exposure is associated with a reduced incidence of dementia when compared with ACEIs or other antihypertensive drugs [9,10]. A recent meta-analysis consisting of 15 observational studies and seven randomised controlled trials (RCT) with 649,790 participants (3.02% of whom had dementia) found that calcium channel blockers (CCBs) and ARBs may significantly reduce the risk of dementia in participants [11]. One study found that although all antihypertensives benefit overall cognition, ARBs may be the most effective [12]. This is supported by another study [13] that found CCBs and ARBs to be superior to other options in preventing cognitive decline and dementia in observational studies. However, this difference was not found in an individual patient data (IPD) meta-analysis study [14]. The IPD study found no difference in effectiveness between drug classes in reducing the risk of dementia [14]. Additionally, three studies in elderly patients with dementia found no significant difference between types of antihypertensive and incidence of cognitive decline [14,15,16].

Despite these limitations, accumulating data in basic science supports a role for ARBs in neuroprotection [17,18,19]. However, no conclusive review is available to suggest the superiority of ARBs in comparison to other antihypertensives in reducing the risk of dementia or preventing cognitive decline. Although systemic reviews on antihypertensive medication classes and dementia have been performed, no review has looked at evidence of associations between ARBs and cognitive function compared to both other antihypertensives and placebo groups. A comprehensive review of relevant clinical studies will identify and evaluate whether ARBs specifically have a positive, negative, or no impact on dementia and whether the effects are superior to those of other antihypertensives. In this systematic review, we aim to comprehensively collate and synthesise current evidence from RCTs to investigate whether ARBs are superior to other antihypertensives and placebos in reducing the onset and progression of cognitive impairment or dementia.

## 2. Methods

The methodologies utilised to report this review are based on the Preferred Reporting Items for Systematic Reviews and Meta-Analyses (PRISMA) statement, which is used for reporting in systematic reviews [20].

### 2.1. Literature Search

Key mesh terms (Table 2) alongside Boolean operators were used to search databases of the Cochrane Database of Systematic Reviews (CDSR), Medical Literature Analysis and Retrieval System Online (MEDLINE), Excerpta Medica dataBASE (Embase), clinicaltrials.gov, accessed on 22 February 2022, and Web of Science. The search was restricted to research conducted in the last 30 years, outlined as follows: CDSR (1992–2022), MEDLINE (1992–2022), Embase (1992–2022), clinicaltrials.gov, accessed on 22 February 2022 (2000–2022), and Web of Science (2000–2022). References were collected using the reference management tool EndNote 20. The final literature search was performed in March 2022.

The population, intervention, comparator, and outcome (PICO) framework was used to construct a search strategy (Table 3). The search was limited further to studies conducted on a human population and published in the English language. The screening and selection procedure for studies used pre-determined inclusion and exclusion criteria (Table 4). Both the titles and abstracts were thoroughly examined, and any studies identified as duplicates were removed.

### 2.2. Data Extraction and Quality Appraisal

Data extraction was performed by one reviewer. Data that were extracted included details of the author(s), year of study, demographic characteristics, blinding of participants (performance bias), blinding of outcome assessors (detection bias), missing outcome data (attrition bias), and selective reporting (reporting bias) [21].

## 3. Results

A literature search using five databases was conducted with the following restrictions: English language, a human population, and studies conducted within the last 30 years (1992–2022). The search produced a total of 1228 references: 97 from Medline, 981 from EMBASE, 72 from CDSR, 7 from clinicaltrials.gov, accessed on 22 February 2022, and 71 from Web of Science. After the exclusion of 154 duplicates, the remaining 1074 references were screened for eligibility; 783 references were excluded following a title and abstract screen, 39 references due to unavailability of full text articles, and 239 references due to failing to meet the inclusion criteria. The screening process, summarised in Figure 1, yielded 13 randomised controlled trials with a total of 26,907 patients which were included in this review [20] (Table 5).

### 3.1. ARB vs. Placebo

Five trials compared ARBs to a placebo. Only one trial showed less deterioration in attention and episodic memory in neurologically stable patients with hypertension. However, no significant difference in cognitive test stores in working memory or executive function was found [27]. In another group of neurologically stable patients, no significant differences in cognitive function were found [28]. In patients presenting with mild-to-moderate AD, there was no significant difference in cognitive function or loss of brain volume [34]. In hypertensive patients with stroke [26] and stroke-elevated events [30], no significant difference was observed in cognitive function. Instead, the results in patients with stroke-related events [30] showed that ARBs produced more harmful effects; however, these were not statistically significant.

### 3.2. ARB vs. ACEI

Two studies compared the effects of ARBs to those of ACEIs. In one study, in patients with mild cognitive impartment (MCI), there was a significant advantage of ARBs compared to ACEIs [32]. Treatment with ARBs was associated with improvement in executive function and episodic memory compared with ACEI treatment. One other study showed that ARBs were associated with higher improvement in scores in word list memory and word list recall compared to ACEIs [24].

### 3.3. ARB vs. CCB

Two studies compared ARBs to CCBs. One study did not show a significant advantage of ARBs compared to CCBs. It should also be noted that although ARBs did not significantly increase cognitive performance scores, the group did show improvement, in comparison to CCBs, in the regional cerebral blood flow (rCBF) in several cerebral sub regions, including the parietal lobe, which is the most severely affected region in AD [29]. Another study showed no significant difference between the test scores [25].

### 3.4. ARB vs. Thiazide

One study compared ARBs to thiazide [22]. Patients randomised to the ARB group showed significant improvement in cognitive test scores compared to the thiazide group.

### 3.5. ARB vs. Beta Blocker

One study compared ARBs to beta-blockers. Those treated with ARBs showed greater improvement in both word list memory and word list recall scores [23].

### 3.6. ARB vs. Statin

Two studies compared ARBs to statins. Although it was shown that statins were better at reducing cognitive decline, combining both statins and ARBs resulted in an even better performance [31,33]. Reduced cognitive decline development and incidence of dementia have also been observed within patients with a combined treatment of ARBs and statins.

### 3.7. Quality Assessment

Table 6 and Table 7 present the results of a quality appraisal using the Critical Appraisal Skills Programme (CASP) checklist and the Cochrane risk of bias tool (RoB2) [21], respectively. The CASP checklists [35] revealed two studies with low risk of bias [23,26], six studies with a moderate risk of bias [22,23,26,27,28], and four studies with a high risk of bias [24,29,30,33]. RoB2 revealed that eight studies had a low risk of bias [24,25,26,27,31,32,33,34], two studies had a high risk of bias [29,30], and three studies had an uncertain risk of bias [22,23,28].

## 4. Discussion

Dementia is currently the seventh leading cause of death, currently affecting more than 55 million people worldwide. This number is expected to increase by 205% by 2050 [36]. Hypertension is associated with an increased risk of dementia, and its treatment may delay the onset of dementia. The aim of this review was to investigate the impact of treatment of hypertension with ARBs on cognitive function and dementia in comparison to other antihypertensives. This review includes data from 13 RCTs. Our findings showed that although ARBs are superior to some antihypertensives such as ACEs, thiazide, and beta blockers, they made no difference in comparison to the placebo. Furthermore, the positive effects on cognitive performance are equal to those of CCBs and lower than those of statins. The neuroprotective effects of ARBs are also more beneficial when they are taken at the same time as statins.

Several studies support our findings that ARBs are superior to ACEs, thiazide, and beta blockers and equal in effectiveness to CCBs. The results also provide suggestive evidence, needing further confirmation, that ARBs are not superior to a placebo. Out of the five RCTS that compared ARBs to placebos, all except one RCT showed no significant influence on cognitive decline [26,27,28,30]. One exception is a study performed by Saxby and colleagues as part of the Study on Cognition and Prognosis in the Elderly (SCOPE) trial. This sub-study of the SCOPE trial recruited 257 elderly (mean age = 76) adults with hypertension. The study (35) found that although the effect sizes were in the small-to-moderate range, the ARB group (8–16 mg of candesartan) showed less decline in attention, episodic memory, and speed of cognition for a mean follow-up period of 44 months. However, the group did not show any difference in working memory or executive function. This was not reflected in the other four studies, where comparisons to the placebo were made and where the patient age groups were 65–86, >55, and mean age 70. Notably, out of the five placebo studies, the SCOPE sub-study had the longest follow-up period, at 44 months, and was the only study in patients with no other comorbidities. It is interesting to note that an additional antihypertensive was permitted for both the ARB and placebo groups to meet the blood pressure target. As the SCOPE trial was the only study out of the five placebo studies with a neurologically stable patient group with no other comorbidities, the neuroprotective effects of ARBs should not be dismissed but rather investigated further.

Eight studies measured cognitive performance and ARB treatment in comparison to other antihypertensives: statins, beta blockers, thiazide, CCBs, and ACEIs [22,23,24,25,29,32,33,35]. The current literature supports the finding that ARBs are preferred over other antihypertensives but work as well as CCBs. Amongst the eight, Zhang et al. and Hu et al. both found an enhanced effect when an ARB was combined with a statin; a synergistic effect was found, with a reduction in white matter lesions and risk of dementia, respectively [31,33]. This may be due to the association between hypertension and dyslipidaemia, and thus the added effect of the statin on plasma lipids may have heightened the effect of telmisartan [37]. Tedesco et al. and Fogari et al. found that losartan, compared to HCTZ or atenolol, respectively, was associated with significant improvements in cognitive tests [22,23]. This finding is supported by a mice study by Barnes et al. where losartan was found to provide an analgesic effect and improved cognitive function without diminishing alertness [34]. Previous studies suggest that this effect is attributed to the overexpression of angiotensin II receptor type 2 (AT2) receptors in the brain, caused by the blocking of ATI receptors by ARBs, resulting in an increase in cerebral blood flow and improved brain metabolism [38,39]. Kehoe et al. found no significant effect of losartan in reducing brain atrophy in patients with mild-to-moderate AD [34]. However, this could be a type II error cause by a small sample size [34].

It is interesting to note, however, that some of these studies showed an increase in cognitive performance from baseline during ARB treatment. This is evident even in studies of varying follow-up periods: 4 weeks, 26 months, 24 months, 16 weeks, and 12 months. Particularly, the comparison of ARBs to thiazide showed that overall cognitive performance improved across all ages (from 30 to 73). However, the longest study of them all showed no cognitive improvement [25]. The study, consisting of 1405 hypertensive patients with a cerebral event in the last 24 months, showed no significant difference in mini-mental state examination (MMSE) scores after 2.5 years of treatment with ARBs and CCBs [25]. Interestingly, not all studies that showed improvement from baseline involved patients without any cognitive deficits. One study in 176 adults with mild cognitive impairments showed that patients on ARBs had improved executive function and memory [32]. These variations in baseline characteristics between RCTs make it difficult to generate direct comparisons on the effect of ARBs on cognitive function. However, all RCTs assessed the effect of ARBs on cognitive function, reported as either a primary or a secondary outcome. Therefore, these findings allow for assumptions on whether ARBs have an impact on cognitive decline in patients both with and without preceding cognitive impairment.

Three studies included patients with hypertension and additional comorbidities within their study populations [25,26,30]: Schrader et al. used a population of hypertensive patients with a history of CVD; the Prevention Regimen For Effectively Avoiding Second Strokes (ProFESS) trial included patients with a recent (90 days) history of an ischaemic stroke and two comorbidities such as hypertension and diabetes; Hornslien et al. studied patients with hypertension and a recent history of stroke [25,26,30]. All three studies showed no significant differences in the mean MMSE scores between their treatment groups. It is unclear whether the presence of comorbidities had an impact on the effectiveness of ARBs. Sayed et al. conducted an experiment on spontaneously hypertensive rats (SHRs) against normotensive rats with a previous history of stroke [40]. Findings from this study show that pre-existing hypertension worsens secondary neurodegeneration following stroke beyond its initial impact on neurovascular damage, suggesting that stroke may impair the effectiveness of ARBs in improving cognition [40]. However, the findings of animal research should be interpreted with care because they are more difficult to extrapolate to humans [41].

Furthermore, in the ProFESS trials, treatment with ARBs was temporarily stopped or ended altogether in patients after a stroke incident, thereby limiting the length of exposure [26]. This may have led to the statistically insignificant MMSE scores because the drug had a limited time to induce any therapeutic effect [26]. However, the author argues that due to the 20 h half-life of the ARB used and the fact that the drug was present in the brain, concerns regarding the therapeutic window are irrelevant.

### 4.1. Strengths and Limitations of RCTs

Ten RCTs used a double-blinded approach where both the investigators and the participants were blinded, minimising any performance bias which could have affected the study outcomes [22,23,26,27,28,30,31,32,33,34,42]. Fogari et al. and Schrader et al. used a prospective randomised open, blinded end-point (PROBE) design, which uses a blinded end-point approach; therefore, only the outcome assessors were blinded to the effects of the intervention [24,25]. Therefore, whilst there is a minimal risk of detection bias, there is a potential risk of performance bias due to participants not being blinded [42]. Kume at al. used an open-label design [29]. The lack of blinding of both participants and assessors leads to a high risk of performance and detection bias [42].

Consistent discrepancies in the baseline qualities of the groups being compared are regarded as being due to selection bias [42]. Randomisation, when carried out correctly, is useful because it minimises selection bias in assigning treatments to patients [42]. Seven studies stated their methods of maintaining allocation concealment: four studies used a computer-based generator sequence, and three used either a return-fax method, web-interface, or a telephone assessment with an automated pin [25,31,32,33]. However, six studies did not disclose in detail if and how allocation concealment was performed, and so it is unclear how randomisation was achieved, thus affecting the reliability of this process [22,23,24,26,27,28,29,30,34].

Furthermore, eight studies used an intention-to-treat (ITT) analysis to assess data [25,26,27,28,31,32,33,34]. The primary goal of an ITT analysis is to preserve the effects of randomisation and minimise selection bias [40]. It also replicates routine patient care, preserves the balance of risk variables across treatment arms at baseline, and keeps study power constant because patients are accounted for until the completion of the trial [43]. Fogari et al., however, used a ‘per-protocol’ (PP) analysis [23]. This is an approach similar to ITT; however, PP does not account for patients who strayed from the protocol, making it difficult to determine whether patients in the distinct treatment groups are still comparable for known and unknown risk factors at the end of the trial [43]. As a result, ITT is preferred, as it retains randomisation, with PP used as a complementary sensitivity analysis [43]. Three studies did not report the type of analysis undertaken, which could lead to potential selection bias [24,29,30].

Additionally, both Kume et al. and Hornslien et al. did not report any missing or incomplete data, which increases the risk of attrition bias [29,30]. The underlying evaluative quality of RCTs is that each treatment group is typically balanced in all aspects [37]. However, many study participants are lost to follow-up [37]. Therefore, attrition occurs when individuals have missing data at a given point which inhibits a comprehensive ITT analysis and introduces bias if the characteristics of those lost to follow-up vary across the randomised groups [44]. However, this loss is only significant if the characteristic is associated with the trial’s outcome measures [44]. Therefore, although Hornslien et al. did not report incomplete data, because the research data in both treatment groups were equal, it is unlikely for attrition bias to have had an effect [30].

Lastly, three studies did not report any adverse events, which could have led to potential bias due to selective reporting [27,29]. This can prove to be detrimental because ineffective and damaging therapies may be adopted into clinical practice if adverse events are not adequately reported [37].

This review has several limitations. Firstly, the research lacked uniformity in numerous areas including study designs, patient characteristics, results, drug-dosage, and methodological quality. This resulted in high heterogeneity, and so it may be inappropriate to combine the results to formulate an overall conclusion. Secondly, only references written in English were considered, which might have introduced language bias. Lastly, the review is based on published studies, hence leading to potential publication bias, which may have influenced the results. Moreover, results with statistical significance are more likely to be reported, and therefore this type of bias typically magnifies empirical consequences.

### 4.2. Comparison to Previous Literature and Implications for Future

There are several observational and cohort studies comparing the efficacy of ARBs vs. ACEIs and exploring the association between ARBs and cognition. Goh et al. used a cohort research design to examine 426,089 people to see whether ARBs lower dementia risk compared to ACEI medications [45]. There was some indication that ARB users had a lower risk of dementia than ACEI users [45]. However, the impact was more pronounced early on and faded afterwards [45]. In contrast, Moran et al. evaluated brain atrophy and cognitive deterioration in non-dementia individuals treated with an ACEI or ARB, supporting the idea that ARBs may be better than ACEIs for brain atrophy reduction [43]. The report says the processes behind this distinct association need further examination [46]. Additionally, the OlmeSartan and Calcium Antagonists Randomised (OSCAR) study confirmed the advantages of ARBs with regard to cognitive function [47]. In an open trial observational study, the OSCAR trial evaluated the effect of eprosartan on cognitive function in 25,745 individuals with uncomplicated hypertension over a period of six months [47]. Early findings of this trial revealed that eprosartan enhanced MMSE scores relative to baseline [47].

Keshri et al. performed a prospective observational trial study to examine the effects of telmisartan on cognitive function in patients with hypertension and dementia [48]. This study found that while telmisartan is better at preventing dementia than a placebo, it is not better than donepezil [48]. However, when used together, telmisartan and donepezil performed better in terms of MMSE scores, with telmisartan functioning as a potentiator [48]. Furthermore, Ho et al. observed lower levels of white matter hyperintensities (WMH) volume and better memory performance in patients using BBB-crossing ARBs such as losartan [49]. List-learning memory performance was also better in patients who used BBB-crossing drugs (ARBs or ACEIs) than in non-users (including normotensive participants) [49]. However, most of these claims are from observational studies, therefore highlighting the lack of RCTs available in assessing the effectiveness of ARBs on cognition and dementia [49].

Analyses have revealed that delaying the development of clinical dementia by only five years might dramatically lower the lifetime risk [50]. Therefore, preventing cognitive decline and dementia by using antihypertensive medications may have a significant benefit for public health services. Dementia presently costs the United Kingdom GBP 34.7 billion annually, with the majority of costs attributed to home-based, long-term care and nursing facilities [51,52]. Patients with dementia and their families presently cover two-thirds of this expense, either via unpaid care or private social care [52]. These possible preventative treatments may allow dementia patients and carers to live more autonomously. Therefore, the findings of this systematic review call for further long-term studies to investigate ARBs’ neuroprotective effects on increasing cognitive function and preventing cognitive decline. A clear study that will be able to distinguish the neuroprotective effects of ARBs on dementia would be beneficial, such as a long-term study with mild dementia patients who are treated with other antihypertensives such as ACEIs, which are then replaced with ARBs for a longer duration.

## 5. Conclusions

This review has investigated the relationship between ARBs and the onset and progression of cognitive impairment and dementia. Due to the inconsistencies in the evidence base, it is not possible to make a firm conclusion on the effectiveness of ARBs on cognition and dementia.

However, this review, along with previous literature, shows promise for the use of ARBs in lowering blood pressure whilst slowing down the development of cognitive impairment. With the prevalence of dementia predicted to increase and considering the negative implications of the disease for both patients and their families, it is imperative that future large-sample RCTs be conducted to ascertain the true effect.

## Figures and Tables

**Figure 1 pharmaceutics-14-02123-f001:**
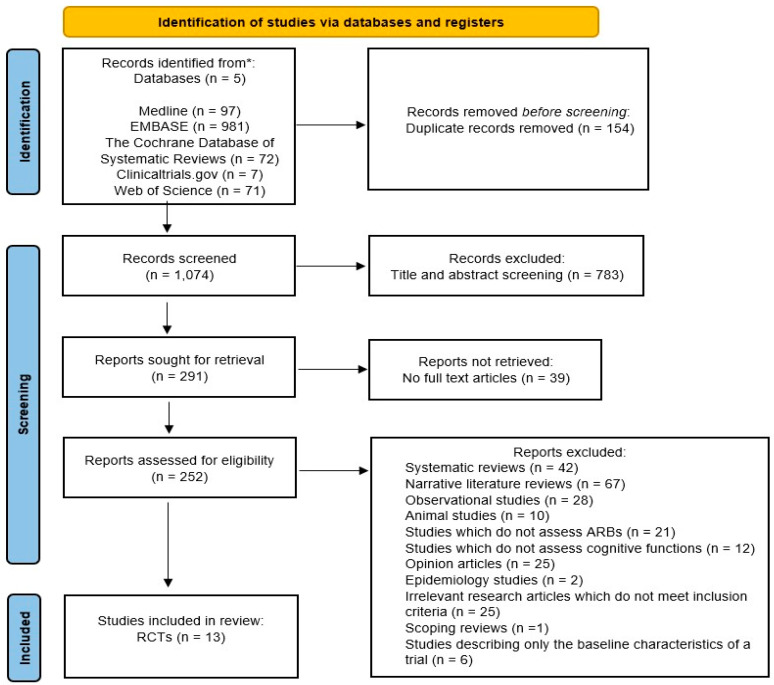
PRISMA diagram. Details of the screening process involved to exclude irrelevant references, which do not meet the inclusion criteria, in order to find appropriate literature to assess the effects of ARBs on cognition or dementia, presented using a PRISMA flowchart [20].

**Table 1 pharmaceutics-14-02123-t001:** The different types of dementia and their clinical characteristics, including disease onset and progression [2].

Type of Dementia	Clinical Features	Onset and Development
**Alzheimer’s disease** This is caused by amyloid (fibrous protein) plaque formation and neurofibrillary tangle formation in the brain.	Progressive memory loss. Communication difficulties. The inability to carry out ordinary tasks. Feelings of anxiety and/or a lack of motivation. A decrease in self-care.	Slow onset. Steady progression over months or years.
**Vascular dementia** Occurs when cerebral blood flow is impeded by vascular disease, causing neuronal malfunction and cell death.	Memory deterioration over time. Communication problems. Chronic deterioration in cognitive performance is accompanied by depression, anxiety, and disinterest.	Can occur in conjunction with a stroke, leading to cognitive impairment. Occurs within minutes or days.
**Lewy bodies** Seen in dementia with Lewy bodies by small aggregates of alpha-synuclein (protein).	Progressive memory loss. Reduced awareness. Confusion. Hallucinations. Falls. Sleep disturbances.	Slow onset. Steady progression over months or years.
**Frontotemporal dementia** Thinking, mood, motivation, and communication are all influenced by the areas of the brain engaged in this process.	Behavioural changes. Speech and language Difficulties. Lowered inhibitions. Semantic dementia.	Slow onset. Steady progression over months or years.
**Mixed dementia** A combination of the most common forms of dementia such as Alzheimer’s disease and vascular dementia.	Similar to Alzheimer’s but alongside strokes or stroke-related events. Vascular risk factors.	More common as age advances (over 80 years).
**Huntington’s disease (HD)** Less prevalent than the others An autosomal dominant hereditary condition.	Causes abnormal movements. Problems with coordination. Dementia is common in around 50% of patients with HD.	Progressive disease which begins mid-life (40–64 years).

**Table 2 pharmaceutics-14-02123-t002:** Key mesh terms applied in CDSR, EMBASE, Medline, Clinicaltrials.gov, accessed on 22 February 2022, and Web of Science to search for appropriate literature.

Database	Search Terms
The Cochrane Database of Systematic Reviews (CDSR)	#1 MeSH descriptor: [Angiotensin Receptor Antagonists] explode all trees OR angiotensin receptor antagonists OR MeSH descriptor: [Losartan] explode all trees OR losartan OR MeSH descriptor: [Telmisartan] explode all trees OR telmisartan OR MeSH descriptor: [Valsartan] explode all trees OR valsartan OR MeSH descriptor: [Irbesartan] explode all trees OR irbesartan OR olmesartan OR candesartan OR eprosartan OR azilsartan
	#2 MeSH descriptor: [Dementia] explode all trees OR dementia OR MeSH descriptor: [Cognitive Dysfunction] explode all trees OR mild cognitive impairment OR cognitive decline
	#3 MeSH descriptor: [Hypertension] explode all trees OR hypertension OR blood pressure
	#4 Combine searches: 1 AND 2 AND 3
	#5 Apply the following limits to search 4: 1992–2022; Human; English Language
EMBASE	#1 Angiotensin II receptor blocker.mp. OR explode angiotensin receptor antagonist/OR explode losartan/OR losar-tan.mp. OR explode losartan potassium/OR explode candesartan/OR explode candesartan hexetil/OR can-desartan.mp. OR explode irbesartan/OR irbesartan.mp. OR explode valsartan/OR valsartan.mp. OR olmesar-tan.mp. OR explode olmesartan/OR telmisartan.mp. OR explode telmisartan/OR eprosartan.mp. OR explode eprosartan/OR explode azilsartan medoxomil/OR azilsartan.mp. OR explode azilsartan/
	#2 explode dementia/OR dementia.mp. OR cognitive decline.mp. OR mild cognitive decline.mp. OR explode mild cognitive impairment/
	#3 Hypertension.mp. OR explode hypertension/OR blood pressure
	#4 Combine searches: 1 AND 2 AND 3
	#5 Apply the following limits to search 4: 1992–2022; Human; English Language
Medline	#1 Angiotensin II receptor block-er.mp. OR explode Angiotensin Receptor Antagonists/OR losartan.mp. OR explode Losartan/OR candesartan.mp. OR irbesartan.mp. OR explode Irbesartan/OR explode Valsartan/OR valsartan.mp OR explode Olmesartan Medoxomil/OR olmesar-tan.mp. OR telmisartan.mp. OR explode Telmisartan/OR eprosar-tan.mp. OR Azilsartan.mp.
	#2 dementia.mp. OR explode Dementia/OR explode Dementia, Vascular/OR cognitive decline.mp. OR explode Cognitive Dysfunction/OR mild cognitive decline.mp. OR mild cognitive impairment/
	#3 Hypertension.mp. OR explode hypertension/OR blood pressure
	#4 Combine searches: 1 AND 2 AND 3
	#5 Apply the following limits to search 4: 1992–2022; Human; English Language
Clinicaltrials.gov, accessed on 22 February 2022	#1 Dementia AND ARB AND cognition AND mild cognitive impairment AND antihypertensives AND angiotensin receptor blockers
	#2 Apply the following limits to search 1: 2000–2022; Human; English Language
Web of Science	#1 Dementia AND hypertension AND angiotensin II receptor blocker AND mild cognitive impairment AND cognitive decline AND antihypertensives AND ARB
	#2 Apply the following limits to search 1: 2000–2022; Human; English Language

**Table 3 pharmaceutics-14-02123-t003:** The PICO framework used to construct the search strategy which details the population, the intervention, the comparator, and the outcome used to ensure that the inclusion criteria follow a specific guideline.

	PICO Framework
Population	Hypertensive patients with or without cognitive deterioration
Intervention	Angiotensin II receptor blockers (ARBs); use of at least one ARB against a comparator in studies using a 2 × 2 factorial design
Comparator	Placebo or other antihypertensives
Outcome	Examine the effect of ARB on cognitive function.

**Table 4 pharmaceutics-14-02123-t004:** Inclusion and exclusion criteria used in the screening process to find relevant references required for the systematic review.

Inclusion Criteria	Exclusion Criteria
Studies conducted within the last 30 years	Follow-up of less than three months
Studies using angiotensin II receptor blockers (ARBs) against placebo or another antihypertensives	Non-hypertensive patients
Antihypertensive patients with mild to moderate hypertension and/or reported increase in blood pressure	Patients suffering from low blood pressure (hypotension)
Any population and any ethnicity	Patients on ARB medication prior to starts of study and no wash-out period applied
Assess effect of specified ARB on cognition as a reported outcome	Studies which only assess effects of ARBs on hypertension and not on cognition
Assesses at least one neuropsychological outcome	Systematic reviews, opinions studies, observational studies including cross-sectional and longitudinal cohort studies, case-studies, narrative literature reviews, conference abstracts, epidemiology studies, animal studies, scoping reviews
Functional cognition assessed using appropriate tests	No access to full-text articles
Randomised controlled trials and prospective randomised open blinded end-point (PROBE) study designs	Studies which do not use ARBs

**Table 5 pharmaceutics-14-02123-t005:** The study characteristics for the included 13 RCTs which detail the population sample, age, patient characteristics, follow-up years, intervention with dosage, outcomes, psychometric tests, study design, and summary of results.

Study	Design	Summary of Results
Tedesco et al. (1999) [22]	Participants: n = 69	Losartan treatment group showed significant improvements in SCAG (*p* < 0.001) and MMSE (*p* < 0.001) scores compared to HCTZ.
	Age: 30–73 years	
	Population characteristics: Patients with uncomplicated mild-to-moderate essential hypertension	
	Study design: Double-blind, randomised controlled trial	
	Groups: (a) ARB (losartan) = 50 mg once daily. (b) Hydrochlorothiazide (HCTZ) (diuretic) = 25 mg once daily.	
	Follow-up: 26 months	
	Psychometric tests: Mini-mental state examination (MMSE), Sandoz Clinical Assessment–Geriatric (SCAG)	
Fogari et al. (2003) [23]	Participants: n = 120	Losartan was better at improving scores in the cognitive tests such as for word list memory (+2.2, *p* < 0.05 vs. baseline) and for word list recall (+2.1, *p* < 0.05 vs. baseline).
	Age: 75–89 years	
	Population characteristics: Patients with uncomplicated mild-to-moderate essential hypertension	
	Study design: Double-blind, randomised controlled trial	
	Groups: (a) ARB (losartan) = 50 mg increased to 100 mg if needed. (b) Beta blocker (BB) (atenolol) = 50 mg increased to 100 mg if needed.	
	Follow-up: 6 months	
	Psychometric tests: Word list memory test, word list recall test, word list fluency test	
Fogari et al. (2004) [24]	Participants: n = 150	Valsartan increased both word list memory (+11.8%; *p* < 0.05 vs. baseline and *p* < 0.01 vs. enalapril) and word list recall scores (+18.7%; *p* < 0.05 vs. baseline and *p* < 0.01 vs. enalapril) compared to enalapril.
	Age: 61–80 years	
	Population characteristics: Patients with uncomplicated mild-to-moderate essential hypertension	
	Study design: PROBE	
	Groups: (a) ARB (valsartan) = 160 mg once daily (b) ACEI (enalapril) = 20 mg once daily	
	Follow-up: 6 months	
	Psychometric tests: Verbal fluency, the Boston naming test, word list memory	
Schrader et al. (2005) [25]	Participants: n = 1405	MMSE scores showed no significant differences in the mean values of the two drugs at the start and end of study.
	Age: 56–85 years	
	Population characteristics: Patients with hypertension and a history of CVA	
	Study design: Prospective, randomised, controlled, and multi-centre study design	
	Groups: (a) ARB (eprosartan) = 600 mg once daily (b) DHP (nitrendipine) = 10 mg once daily	
	Follow-up: 30 months (mean)	
	Psychometric tests: MMSE	
Diener et al. (2008) [26]	Participants: n = 20,332	No significant differences in median MMSE scores.
	Age: 66 years (mean)	
	Population characteristics: Patients with an ischemic stroke in the last 90 days or 90–120 days after stroke. Additional risk factors such as diabetes, hypertension, etc.	
	Study design: 2 × 2 factorial, double blind, randomised controlled trial	
	Groups: (a) ARB (telmisartan) = 80 mg once daily (b) Placebo	
	Follow-up: 28.8 months (mean)	
	Psychometric tests: MMSE	
Saxby et al. (2008) [27]	Participants: n = 257	Candesartan group showed less deterioration in attention compared to placebo (0.004 vs. 0.036, *p* = 0.0428) and for episodic memory (0.14 vs. 0.22, *p* = 0.0428). No differences were found between the two groups for working memory
	Age: 70–89 years	
	Population characteristics: Patients with hypertension and an MMSE score of 24 and above.	
	Study design: Placebo-controlled, double-blind, randomised controlled trial	
	Groups: (a) ARB (candesartan) = 8 mg once daily increased to 16 mg if needed (b) Placebo	
	Follow-up: 44 months (mean)	
	Psychometric tests: Cognitive drug research computerised assessment battery, trail-making test (TMT) and verbal fluency tests	
Flesch et al. (2009) [28]	Participants: n= 112	1 week and 3 months after follow-up, there were no differences in CABG compared to before start of surgery.
	Age: 65–85 years	
	Population characteristics: Patients undergoing cognitive function post- surgery (CABG) with essential arterial hypertension and neurologically stable	
	Study design: Randomised, double-blind, placebo-controlled study	
	Groups: (a) ARB (candesartan) = 8 mg once daily (b) Placebo	
	Follow-up: 3 months	
	Psychometric tests: Zimmerman’s divided attention test, Reitan’s trail making test version A and B and two questionnaires.	
Kume et al. (2012) [29]	Participants: n = 20	CCB (amlodipine) produced lower scores in the neurophysiological tests, whereas ARB (telmisartan) did not show any changes. Telmisartan showed increased regional cerebral blood flow (rCBF) in several parts of the brain compared to amlodipine.
	Age: >70 years	
	Population characteristics: Patients with mild Alzheimer’s disease (AD) with essential hypertension	
	Study design: Prospective randomised, open-label, parallel design	
	Groups: (a) ARB (telmisartan) = 40 mg daily increased to 80 mg if needed (b) CCB (amlodipine) = 5 mg daily increased to 10 mg if needed.	
	Follow-up: 6 months	
	Psychometric tests: MMSE, Alzheimer’s Disease Assessment ScaleCognitive Subscale Japanese version (ADAS-Jcog), Wechsler Memory Scale (WMS-R).	
Hornslien et al. (2013) [30]	Participants: n = 2029	No significant difference in the distribution of the MMSE scores between the two groups (95% CI of 0.91–1.34). Candesartan also showed signs of harmful effects instead of improving cognitive function; however, this is not statistically significant enough to report as harmful.
	Age: ≥18 years	
	Population characteristics: Patients with a stroke-related event within 30 h of presenting and with a SBP ≥ 140 mmHg	
	Study design: Multicentre, randomised controlled, placebo-controlled, double-blind trial	
	Groups: (a) ARB (candesartan) = 4 mg once daily increased to 16mg if needed (b) Placebo.	
	Follow-up: 6 months	
	Psychometric tests: MMSE	
Zhang et al. (2019) [31]	Participants: n = 732	No significant differences in white matter hyperintensities (WMH) development and cognitive deterioration between the two treatment groups. Rosuvastatin associated with lower risks of cognitive impairment (95% CI: 0.36–0.80). Telmisartan was somewhat more effective than a comparable placebo in slowing the course of WMH and cognitive impairment. Telmisartan and rosuvastatin worked better together to reduce WMH development and improve cognitive decline.
	Age: ≥60 years	
	Population characteristics: Patients with hypertension with SBP ≥ 140 mmHg and DBP ≥ 90 mmHg	
	Study design: 2 × 2 factorial, double-blind, randomised controlled design	
	Groups: (a) ARB (telmisartan) = 40 mg once daily increased to 80 mg if needed and a placebo drug (b) Low-dose statin (rosuvastatin) = 10 mg once daily and a placebo drug.	
	Follow-up: 59.8 months (mean)	
	Psychometric tests: MMSE, Dementia Rating Scale (DRS)	
Hajjar et al. (2020) [32]	Participants: n = 176	Candesartan group performed better in the trail making test part B but not in the executive abilities tool exam. Candesartan group also performed better in the HVLT-R test and on retention compared to lisinopril group.
	Age: ≥55 years	
	Population characteristics: Patients with a history of hypertension and MCI of the executive or mixed type	
	Study design: Investigator-initiated, single-centre, double-blind, randomised controlled trial.	
	Groups: (a) ARB (candesartan) = 8 mg once daily increased to 16 mg to 32 mg if needed (b) ACEI (lisinopril) = 10 mg once daily increased to 20 mg to 40 mg if needed.	
	Follow-up: 12 months	
	Psychometric tests: TMT (A&B), Measures and Instruments for Neurobehavioral Evaluation and Research toolbox, Hopkins Verbal Learning Test-Revised (HVLT-R), The Boston Naming Test	
Hu et al. (2020) [33]	Participants: n =1244	Both telmisartan and rosuvastatin reduced cognitive decline development and the incidence of dementia. A synergistic effect was found when the two drugs worked together to reduce cognitive decline.
	Age: ≥60 years	
	Population characteristics: Patients with essential hypertension and neurologically stable	
	Study design: Double-blind, randomised, placebo-controlled trial using a 2 × 2 factorial design	
	Groups: (a) ARB activator (telmisartan) = 40 mg once daily increased to 80 mg if needed and a statin (rosuvastatin) placebo (b) ARB placebo, rosuvastatin placebo (c) ARB placebo and rosuvastatin activator = 10 mg once daily (d) ARB activator = 40 mg once daily increased to 80 mg if needed and a rosuvastatin activator = 10 mg once daily.	
	Follow-up: 84 months (mean)	
	Psychometric tests: MMSE, Montreal Cognitive Assessment (MoCA), DRS, clinical dementia rating (CDR)	
Kehoe et al. (2021) [34]	Participants: n = 261	Losartan was well tolerated; however, it was not effective in reducing brain degeneration in patients.
	Age: ≥55 years	
	Population characteristics: Patients with mild-to-moderate AD and with or without hypertension	
	Study design: Parallel, two-arm, double-blind, placebo	
	Groups: (a) ARB (losartan) = 25 mg once daily increased to 100 mg.	
	Follow-up: 12 months	
	Psychometric tests: ADAS-Cog, The Neuropsychiatry Inventory	

**Table 6 pharmaceutics-14-02123-t006:** The results of the CASP tool [20], which are used to guide the reviewers to answer specific questions about the study design in order to determine the quality of the study.

CASP Checklist	1 [22]	2 [23]	3 [24]	4 [25]	5 [26]	6 [27]	7 [28]	8 [29]	9 [30]	10 [31]	11 [32]	12 [33]	13 [34]
Did the study address a clearly focused research question?	Y	Y	Y	Y	Y	Y	Y	Y	Y	Y	Y	Y	Y
Was the assignment of participants to interventions randomised?	UC	UC	N	Y	UC	Y	UC	N	Y	Y	Y	Y	Y
Were all participants who entered the study accounted for at its conclusion?	N	Y	N	Y	Y	Y	Y	N	N	Y	Y	Y	Y
What type of blinding took place?	DB	DB	EPB	EPB	DB	DB	DB	OL	DB	DB	DB	DB	DB
Were the study groups similar at the start of the randomised controlled trial?	Y	Y	Y	Y	Y	Y	Y	Y	Y	Y	Y	N	Y
Apart from the experimental intervention, did each study group receive the same level of care (that is, were they treated equally)?	Y	Y	Y	Y	Y	Y	Y	Y	Y	Y	Y	Y	Y
Were the effects of intervention reported comprehensively?	Y	Y	Y	Y	Y	Y	Y	Y	Y	Y	Y	Y	Y
Was the precision of the estimate of the intervention or treatment effect reported?	Y	Y	Y	Y	Y	N	N	N	Y	Y	Y	Y	Y
Do the benefits of the experimental intervention outweigh the harms and costs?	UC	UC	UC	UC	UC	UC	UC	UC	UC	UC	UC	UC	UC

Y, Yes; UC, Unclear; N, No; DB, Double-blind; EPB, End-point-blind; OP, Open-label.

**Table 7 pharmaceutics-14-02123-t007:** The Cochrane RoB2 checklist [21], used to assess potential biases in the included randomised controlled trials.

Author and Year of Study	Cochrane RoB2 Checklist						
	Random Sequence Generation (Selection Bias)	Allocation Concealment (Selection Bias)	Blinding of Participants (Performance bias)	Blinding of Outcome Assessors (Detection Bias)	Incomplete Outcome Data (Attrition Bias)	Selective Reporting of Adverse Events (Reporting Bias)	Overall RoB2
Tedesco et al. (1999) [22]	UCR	UCR	LR	LR	LR	LR	UCR
Fogari et al. (2003) [23]	UCR	UCR	LR	UCR	LR	LR	UCR
Fogari et al. (2004) [24]	UCR	HR	HR	LR	LR	LR	LR
Schrader et al. (2005) [25]	LR	UCR	HR	LR	LR	LR	LR
Diener et al. (2008) [26]	UCR	UCR	LR	UCR	LR	LR	LR
Saxby et al. (2008) [27]	LR	UCR	LR	UCR	LR	HR	LR
Flesch et al. (2009) [28]	UCR	UCR	LR	UCR	LR	LR	UCR
Kume et al. (2012) [29]	UCR	UCR	HR	UCR	HR	HR	HR
Hornslien et al. (2013) [30]	LR	UCR	LR	LR	HR	HR	HR
Zhang et al. (2019) [31]	LR	UCR	LR	LR	LR	LR	LR
Hajjar et al. (2020) [32]	LR	UCR	LR	UCR	LR	LR	LR
Hu et al. (2020) [33]	LR	UCR	LR	LR	LR	LR	LR
Kehoe et al. (2021) [34]	LR	UCR	LR	LR	LR	LR	LR

UCR, unclear risk; LR, low risk; HR, high risk.

## Data Availability

Not applicable.

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
