# Peer review of "Angiotensin II Receptor Blockers in the Management of Hypertension in Preventing Cognitive Impairment and Dementia—A Systematic Review"

_pharmaceutics, 2022, doi:10.3390/pharmaceutics14102123_

Round 1

Reviewer 1 Report

1. Some grammatical and punctuation errors need to be corrected in the article.

2. The authors suggested that ARBs were more effective than other antihypertensives but had a similar effect to placebo. It does not seem logical that they are effective from other drugs and not from placebo. Authors should clarify the issue.

Author Response

Thank you for reviewing our manuscript, " Angiotensin II receptor blockers in the management of hypertension in preventing cognitive impairment and dementia – a systematic review" (Manuscript ID pharmaceutics-1914218). We have addressed your comments and the comments from Reviewer #1, and #2 below. Original comments are in red, and our reply is in bold, and the changes in the main text have been highlighted in yellow.

-Reviewer 1

  1. Some grammatical and punctuation errors need to be corrected in the article.

Response: Many thanks for the comment. We have checked the grammar and punctuation and corrected the necessary errors.

  1. The authors suggested that ARBs were more effective than other antihypertensives but had a similar effect to placebo. It does not seem logical that they are effective from other drugs and not from placebo. Authors should clarify the issue.

Response: Many thanks for the comment. This issue has now been clarified in the discussion section.

Reviewer 2 Report

This manuscript belongs to the long range of papers dealing with possible healing effect of  ARBs  on dementia. The most recent ones are Deng et al. 2022 and Yasar and Wharton 2022. 

The authors have to clearly emphasises in what aspect is their work exceptional.

The Introduction is short. It should contain what is known from observational studies about ARBs and dementia. Only one sentence is devoted to this issue: "Despite these limitations, accumulating data in basic science supports a role for ARBs 70 in neuroprotection [13-15].". If the observational studies show significant effect of ARBs on dementia the authors have to discuss the possible mechanism in Introduction or Discussion. 

In addition, the authors have to suggest what kind of study or clinical trial would be clearly distinguish whether the ARBs have or not have positive impact on dementia. For exmple to suggest clinical trial with mild dementia patient treated with ACEi and then replaced with ARBs for longer time.

Explanation of some abbreviations is missing. 

Author Response

Thank you for reviewing our manuscript, " Angiotensin II receptor blockers in the management of hypertension in preventing cognitive impairment and dementia – a systematic review" (Manuscript ID pharmaceutics-1914218). We have addressed your comments and the comments from Reviewer #1, and #2 below. Original comments are in red, and our reply is in bold, and the changes in the main text have been highlighted in yellow.

-Reviewer 2

This manuscript belongs to the long range of papers dealing with possible healing effect of ARBs on dementia. The most recent ones are Deng et al. 2022 and Yasar and Wharton 2022.

The authors have to clearly emphasises in what aspect is their work exceptional.

Response: Many thanks for the comment. We have added a sentence on how our work is exceptional.

The Introduction is short. It should contain what is known from observational studies about ARBs and dementia. Only one sentence is devoted to this issue: "Despite these limitations, accumulating data in basic science supports a role for ARBs 70 in neuroprotection [13-15].". If the observational studies show significant effect of ARBs on dementia the authors have to discuss the possible mechanism in Introduction or Discussion.

Response: Many thanks for the comment. Observational studies have been included and the possible mechanisms has been added to the introduction.

In addition, the authors have to suggest what kind of study or clinical trial would be clearly distinguish whether the ARBs have or not have positive impact on dementia. For exmple to suggest clinical trial with mild dementia patient treated with ACEi and then replaced with ARBs for longer time.

Response: Many thanks for the comment. We have explained what kind of studies would aid the current understanding of the impact of ARBs on dementia.

Explanation of some abbreviations is missing.

Response: Many thanks for the comment. We have checked the entire manuscript for missing abbreviations and inserted them where they were previously missing.

Round 2

Reviewer 2 Report

The manuscript is significantly improved.

Minor comments:

Line 64: renin acts on angiotensinogen (not angiotensin)

Line 239 "bur" has to be replaced by "but"

Author Response

Thank you for reviewing our manuscript, " Angiotensin II receptor blockers in the management of hypertension in preventing cognitive impairment and dementia – a systematic review" (Manuscript ID pharmaceutics-1914218R2). We have addressed your comments and the comments from Reviewer #2 below. Original comments are in red, and our reply is in bold, and the changes in the main text have been highlighted in yellow.

We have carefully checked through the manuscript for many times, to hopefully avoid any mistakes or typos etc..

Comments from the editors and reviewers:

-Editors

Please improve your manuscript according to the reviewers comments. Pay special attention showing the novelty/exceptional of your work. Please write also whether there is any other mechanism of neuroprotective action of angiotensin II inhibitor independent of normalization of hypertension.

Response: Many thanks for the comment. We have added some sentences on other mechanisms of neuroprotective action of angiotensin II inhibitor independent of normalization of hypertension to the text.

-Reviewer 2

The manuscript is significantly improved.

Response: Many thanks for the comment.

Minor comments:

Line 64: renin acts on angiotensinogen (not angiotensin)

Response: Many thanks for the comment. We have changed angiotensin to angiotensinogen in the text.

Line 239 "bur" has to be replaced by "but"

Response: Many thanks for the comment. We have changed “bur” to “but” in the text.